# A New Approach to the Determination of Silicon in Zinc, Lead-Bearing Materials and in Waste Using the ICP-OES Method

**DOI:** 10.3390/molecules27103059

**Published:** 2022-05-10

**Authors:** Artur Przybyła, Joanna Kuc, Zbigniew Wzorek

**Affiliations:** 1Mining and Metallurgical Works ZGH “Bolesław”, Kolejowa 37, 32-332 Bukowno, Poland; 2Department of Chemical Engineering and Technology, Cracow University of Technology, Warszawska 24, 51-155 Cracow, Poland; joanna.kuc@pk.edu.pl (J.K.); zbigniew.wzorek@pk.edu.pl (Z.W.)

**Keywords:** silicon, ICP-OES, zinc and lead concentrates, waste

## Abstract

This work describes the implementation of the ICP-OES silicon determination method for zinc and lead-bearing materials and waste at the Mining and Metallurgical Works ZGH “Bolesław”. The proposed method was validated. On the basis of linearity tests, it was found that the course of the calibration curve is linear up to a silicon concentration of 100 mg/L, with the determined working range being 0.10–50%. Precision tests, on the basis of which the repeatability was checked, were carried out for nine types of real samples: zinc sulfides, zinc oxides, zinc-lead ore, lead sulfide and zinc-bearing waste. Real samples and six certified reference materials were tested using the ICP-OES radial position. The identified interferences of molybdenum, chromium and vanadium did not statistically significantly affect the measurement results.

## 1. Introduction

Silicon is ubiquitous in the lithosphere and biosphere of the Earth. It is the second most abundant element, after oxygen, in the Earth’s crust and accounts for over 25% of its mass, most often in the form of silicate minerals [1]. At room temperature, silicon is chemically inactive, due to the fact that the surface of its crystallites is always covered with a thin layer of silicon dioxide [2]. In most research laboratories, silicon is determined by the spectrophotometric method after mineralization in perchloric acid, followed by melting the sample in potassium hydroxide and measuring the yellow color as a result of the reaction of dissociated silicon with ammonium molybdate. This method uses explosive perchloric acid, which is hazardous (mineralization in an open system) due to the possibility of perchlorate crystallization in ventilation ducts [2]. For this reason, it is necessary to develop new methods for the determination of silicon that exclude the use of perchloric acid. 

Owing to the specificity of the studied objects (zinc and lead concentrates, zinc-lead ores), the literature on the determination of silicon under the aforementioned conditions is limited. Several companies in Europe produce electrolytic zinc. The Glencore group produces approx. 800,000 tonnes of zinc per year, the Nyrstar group produces approx. 600,000 tonnes per year, the New Boliden group produces approx. 500,000 tonnes, KCM Chelyabinsk produces approx. 200,000 tonnes per year, ZGH "Bolesław" produces about 160,000 tonnes per year and KCM Plovwdivw produces approx. 80,000 tonnes per year.These data provide an overview of the scale of production of zinc concentrates in the European market. They were obtained from the Chief Technologist of “ZGH Bolesław”.

The silicon content of the above-mentioned materials must be monitored. Excess silicon is disadvantageous both in the fluidized-bed roasting process—where it results in sintering and the formation of low-melting silicates—and in the sedimentation process after neutral leaching, because colloidal silica hinders the sedimentation of the solid phase after leaching.

The primary method of determining silicon in the above-mentioned materials is the weight method. In this method, a sample is dissolved in hydrochloric, nitric (V), sulfuric (VI) and hydrofluoric acids and the residue is melted with a flux (sodium-potassium carbonate and sodium tetraborate mixed in a ratio of 3:2 by weight). The acid precipitate is then separated from the combined solutions in the presence of gelatin and the silica is stripped with hydrofluoric acid. The silica content can then be determined from the difference in the mass of the sediment before and after stripping [3]. However, this method is laborious and time-consuming. Several procedures have been proposed for the determination of silicon in various materials using molybdenum blue with UV-VIS spectrophotometry, flame atomization atomic absorption spectrometry or inductively induced plasma atomic emission spectrometry [4,5,6,7,8]. The UV-VIS spectrophotometry method is the standard analytical method [9] and has been approved by the Association of Analytical Chemists (AOAC) [8]. More and more often, for the determination of silicon, spectroscopic methods are used after digesting samples. Digestion in open vessels is the most common approach for the preparation of samples for the determination of silicon by spectrochemical methods [10,11]. However, given the large amounts of reagents required and potential sources of contamination, alternatives have been sought for many years. An example of an alternative sample preparation method is closed vessel microwave irradiation. This method allows for a more controlled mineralization environment, which minimizes contamination and loss of the analyte while ensuring better precision [12,13]. Microwave mineralization is the simplified name for the microwave assisted mineralization process. Heating with microwaves is more effective than conventional heating, because microwave energy is transferred directly to all molecules of the solution almost simultaneously without heating the vessel [14]. Microwave energy, in combination with hydrofluoric and nitric acid, allows for the complete decomposition of sparingly soluble silicon bonds in zinc and lead-bearing materials and the transfer of silicon to the liquid phase in a dissociated form.

ICP optical emission spectrometry [15,16,17,18,19], AAS atomic absorption spectrometry [20,21,22,23,24,25], X-ray XRF and fluorescence [26] are increasingly widely used to determine silicon in various materials. The ICP-OES or ICP-AES technique (the same technique; different abbreviations are given by the authors) measurement parameters include the selection of the analytical line, selection of plasma observation conditions, gas flow rate, radio frequency generator power, photomultiplier voltage, integration time and the amount of sample to be administered. ICP-OES is used for the qualitative and quantitative determination of over 70 elements in a wide range of concentrations, i.e., from 1 ng/L to 1 g/L. High-frequency radio waves are used for atomization and excitation, which enable the production of high-temperature plasma. On this basis, chemical compounds decay into atoms which are then excited, emitting absorbed energy in the form of electromagnetic radiation which is characteristic of the elements present in the sample. This technique is used to analyze the composition of many types of liquid and solid samples after being brought into solution. The temperature distribution in the plasma ranges from 5000 K outside to 10,000 K in the center. Such high temperatures ensure high excitation and ionization efficiency, as well as low chemical interference [27]. Each determination performed with the ICP-OES technique consists of several stages, including the elimination of interference. An interferent is any factor that causes a change in the signal of an analyte in the tested sample in relation to the signal corresponding to the same concentration in a standard solution. The effects include spectral, physical, chemical and ionization interference, with the former being the most common in the ICP-OES technique [27]. Although the ICP-OES technique is well known and routinely used for the determination of many elements, to date, no methodology for the determination of silicon in zinc and lead-bearing materials has been developed using this technique in combination with mineralization with hydrofluoric and nitric acids. For this reason, a new and safe ICP-OES technique combined with microwave mineralization for the determination of silicon in zinc, lead-bearing materials and waste was implemented and validated.

## 2. Results

### 2.1. Mineralization and ICP-OES Measurement Parameters

Appropriate conditions were established by modifying the process several times, including extending the time to reach the temperature of 190 °C. The optimal program for the mineralization of zinc and lead-bearing materials, whereby the time and temperature do not interrupt the mineralization process, is as follows:25 min to reach 190 °C10 min to reach 220 °C15 min for mineralization at 220 °C40 min for cooling

Measurements on an ICP-OES spectrometer were carried out with the parameters listed in Table 1. These parameters were selected based on the manufacturer’s instructions and the authors’ own experience.

### 2.2. Radial and Axial ICP-OES Systems

The zinc and lead-bearing materials used contained silicon in the range 0.1–50%; this operating range was adopted for further calculations. A comparison of the coefficients of variation, limits of quantification and selecitivity obtained in the radial and axial system for each of the analytical lines is presented in Table 2.

### 2.3. Spectral Interference

Figure 1 shows the ICP-OES spectrum of the certified CRM4 reference material enriched with 20% of molybdenum, vanadium and chromium, respectively.

### 2.4. Validation Parameters

A list of the main parameters characterizing the method is given in Table 3.

Figure 2 and Figure 3 show the share of the components in the uncertainty budget for silicon contents of 0.1–10% and 10–50%, respectivly.

Figure 4 shows a graph of the expanded uncertainty as a function of silicon concentration.

### 2.5. Silicon Content in Certified Reference Materials and Test Samples

Table 4 compares the results regarding silicon content obtained using the developed method in certified reference materials with the values provided by the manufacturer. Table 5 summarizes the silicon content determined in the tested samples. The results are given taking into account the value of expanded uncertainty U(x). 

## 3. Discussion

The standard method for the determination of silicon in zinc and lead concentrates and in waste, as recommended by the Association of Analytical Chemists (AOAC), is based on spectrophotometric measurements of UV-VIS spectrophotometry. Due to the interference caused by matrix components, newer methodological solutions are being sought. As such, an analytical methodology using ICP-OES was developed and the sample mineralization process was modified. 

The standard mineralization method in an open system uses explosive perchloric acid. In this work, microwave mineralization with hydrofluoric and nitric acids was used. Nitric acid is the most versatile reagent in sample digestion processes and the most widely used primary oxidant. Hydrofluoric acid breaks the strong bonds of silicate ions and transfers the silicon to the liquid phase in a dissociated form. Additionally, it does not have such a high risk of explosion as perchloric acid (as a result of perchlorate crystallization in ventilation ducts). Hence, hydrofluoric and nitric acids are often used for the mineralization of minerals, ores, soils and rocks containing silicates. 

Increasing the heating time in the mineralization process resulted in the establishment of the optimal temperature program (25 min to 190 °C and 10 min to 220 °C), in which the process runs properly and does not automatically stop. 

The silicon calibration curves in the radial ICP-OES measurements for the analytical lines at 212.412 nm, 251.611 nm and 288.158 nm are linear (r = 1).

The precision of the method for both the radial and axial positions is comparable, except for the 251.611 nm line. The limits of quantification for the radial and axial positions meet the basic validation condition (i.e., they are less than 0.1%). However, for measurements in the axial position, the precision of the method deteriorates (i.e., there is a greater coefficient of variation) and the limit of quantification increases with increased sensitivity. Moreover, the sensitivity of the method for the axial position is about 10 times higher than for the radial position for each analytical line. For these reasons, real samples were tested using the ICP-OES radial position.

The ICP-OES technique allows the measurement of silicon with appropriate precision and gives correct results using a glass mist chamber and a “Mira Mist” type sprayer made of plastic. Possible disturbance related to interference from the glass parts of the ICP-OES instrument did not have a significant impact on the measurement results, as confirmed by analyses of certified materials. 

Differences between the results obtained and the values provided by the manufacturer were not statistically significant. This proves the correctness of the developed analytical procedure. 

Measurements of silicon levels by means of the ICP-OES spectrometer are not free from spectral interferences. According to the literature data [28], for the 212.412 nm analytical line, the interferent is molybdenum; for the 288.158 nm analytical line, it is chromium; the 251.611 nm analytical line is free from spectral interferences. Using the interference preview in the iTeva software, it was noticed that for the 288.158 nm line, vanadium may be an interferent. The effect of chromium can be excluded by disregarding the analytical background to the right of the peak of the spectrum. Molybdenum and vanadium interference can increase the peak area of silicon; however, as proven by testing certified reference materials, these interferences do not have a statistically significant impact on the results of silicon measurements. 

For the range of 0.1–50% silicon, the influence of components related to mass, volume and calibration is small (<10%) (Figure 2 and Figure 3), and therefore, these components can be ignored in the uncertainty budget. The estimated expanded uncertainty meets the criteria assumed in the study, so the method is suitable for the measured application. 

The measurement results obtained with the developed method for certified reference materials were compared with the silicon content declared by the manufacturer. The differences in values are comparable in terms of the determined expanded uncertainty. This indicates the reliability of the proposed method.

In addition, analyses of materials which are routinely tested in analytical laboratories were performed. The highest concentrations of silicon were determined in zinc-lead ore and zinc-bearing waste (25.34 ± 1.29 and 12.11 ± 0.79, respectively). The lowest silicon contents were found in ZnO steel dust and PbS galena materials (0.190 ± 0.051 and 0.310 ± 0.070, respectively). These results correlate with those obtained over many years of laboratory practice at the “Bolesław” ZGH.

## 4. Materials and Methods

### 4.1. Reagents

AccuStandards commercial standard silicon, concentrated hydrochloric acid, concentrated nitric acid, concentrated hydrofluoric acid and concentrated hydrogen dioxide were purchased from MERCK Emsure (Darmstadt, Germany). Ammonium molybdate was supplied by MERCK Supelco (Darmstadt, Germany). 

### 4.2. Research Material

The sampling was carried out in accordance with the principles in ISO 12743: 2018 [29].

The following certified reference materials were used:**CRM_1**—Zinc concentrate (zinc sulfide) with a silicon content of 0.122%—produced by the Institute of Non-Ferrous Metals, Gliwice, Poland;**CRM_2**—Zinc concentrate (zinc sulfide) with a silicon content of 0.295%—produced by the Canadian Certified Reference Materials Project, Ottawa, Canada;**CRM_3**—Lead concentrate (lead sulfide) with a silicon content of 0.305%—produced by the Canadian Certified Reference Materials Project, Ottawa, Canada;**CRM_4**—Zinc oxide with a silicon content of 2.56%—produced by the Institute of Non-Ferrous Metals, Gliwice, Poland;**CRM_5**—Zinc concentrate (zinc sulfide) with a silicon content of 9.30%—manufactured by the China National Analysis Center for Iron and Steel, Beijing, China;**CRM_6**—Zinc ore with a silicon content 38.77%—manufactured by the China National Analysis Center for Iron and Steel, Beijing, China.

A description of the tested samples is presented in Table 6.

### 4.3. Apparatus

The samples were digested in an ETHOS microwave mineralizer, manufactured by MILESTONE. Silicon was determined using an ICP-OES iCAP7400 spectrometer, manufactured by THERMO Scientific (Waltham, MA, USA) and provided with a computer with a control and data collection program (iTeva), which enabled measurements in the radial and axial versions. The ICP-OES instrument allowed us to measure silicon with appropriate precision using a glass mist chamber and a "Mira Mist" type sprayer made of plastic. 

### 4.4. Selecting the Mineralization Parameters

In preparing the method validation plan, it was assumed that the maximum sample weight proposed for mineralization, viz. which could be safely mineralized, would be 0.25 g, and after mineralization, the sample would be transferred to a 250 mL polypropylene volumetric flask. These assumptions were based on the necessity to achieve the best possible precision, given that accuracy deteriorates as the weight of the sample is lowered or the volume of the flask into which the mineralizate is transferred is increased. The first stage of the experiment was to select appropriate mineralization conditions. Using the mineralizer operating instructions and the manufacturer’s application, the following reagents were selected: hydrochloric acid, nitric acid, hydrogen peroxide and hydrofluoric acid. Due to the high boiling point of sulfuric acid and the explosiveness of perchloric acid, a new method was developed to exclude the use of the latter, thereby improving the safety of the method. The time and temperature of the mineralization, as well as the cooling time of the device, remained unchanged, i.e., 15 min at 220 °C and 40 min, respectively. Therefore, the selected parameters constituted the temperature program of the initial stage until 220 °C was reached.

### 4.5. Calculation Methods

Calculations were made using the statistical tools offered in the e-stat package, authorized by the Faculty of Chemistry of the University of Warsaw [30].

The percentage of silicon (X) was calculated according to the formula:X = (b∙V)/(m⋅10,000),(1)
where b is the silicon concentration read from the calibration curve in mg/L; V is the volume of the sample solution in mL; and m is the sample weight in g.

The silicon concentration was determined from the calibration curve and calculated as the arithmetic mean of the silicon concentrations read from three analytical lines.

The mean of at least two parallel determinations was used, between which the difference did not exceed: 15% of the lower result for a silicon content of 0.1–0.5%;10% of the lower result for a silicon content of 0.51–2%;5% of the result lower for a silicon content of 2.01–10%;2.5% of the lower result for a silicon content above 10%.

#### 4.5.1. Working Range

The working range of the method was defined, within which linearity as well as acceptable correctness and precision can be achieved [31]. Within the assumed working range, the homogeneity of variance was checked using the F-Snedecor test. The next step was to select the analytical lines for which the calibration would be performed, and to select the measurement position, i.e., axial or radial. The choice of analytical lines was dictated by several factors: lack of spectral interferents, the presence of which could falsify the measurement results; the high sensitivity of the analytical lines used; a limit of quantification not greater than 0.1% for each of the analytical lines; and precision of the calibration curve not greater than 3% for each of the analytical lines. Based on the literature data [28], the following analytical lines were selected: 212.412 nm, 251.611 nm and 288.158 nm. For the 212.412 nm analytical line, the interferent is molybdenum; for the 288.158 nm analytical line—chromium; the 251.611 nm analytical line is free of spectral interference.

#### 4.5.2. Linearity

The measures of the linearity of the standard curve are the correlation coefficient and residual analysis [31]. In order to determine the linearity of the standard curve, the r-Pearson linear correlation coefficient was determined.

#### 4.5.3. Resistance

The assessment of method resistance consisted of identifying steps in the procedure whose change may have an impact on the result [32]. In the case of the present research, these may have been the type of reagents used for mineralization, the amounts of reagents used for mineralization, the temperature and time of mineralization, the mass of the sample subjected to mineralization and the volume of the flask in which the mineralizate was placed.

#### 4.5.4. Limit of Detection (LOD) and Limit of Quantification (LOQ)

In the present research, two methods for determining the limit of detection and quantification were used: the statistical parameters of the calibration curve [33] and the real sample with a known silicon content close to the beginning of the assumed measuring range of the method.

#### 4.5.5. Selectivity

The selectivity of the method is related to the slope of the lines resulting from the regression analysis. Selectivity is the degree to which other substances in a sample influence the analytical signal [31]. The selectivity of an analytical method is defined as the possibility of determining one component or a group of components in relation to others in a complex real sample, without interference of accompanying components. The method is therefore ideally selective (specific) when, in a complex mixture, the signal is generated only by the analyte. In order to confirm the presence of interferents, a certified material enriched with molybdenum, chromium and vanadium, respectively, was prepared.

#### 4.5.6. Precision

Precision describes the degree of consistency among independent test results for the same sample obtained according to the aforementioned procedures [31]. It is usually expressed as a relative standard deviation (RSD). In order to determine the degree of precision, nine real samples were analyzed under the conditions [34] of within-laboratory reproducibility by carrying out eight series of repetitions. The silicon contents were selected to be close to the upper and lower applicability limits of the method.

#### 4.5.7. Correctness

The parameter was defined as the degree of consistency between the average value of a large number of individual test results and the actual value or the adopted reference value. The correctness of the silicon determinations was checked by comparing the obtained results with the nominal value of certified reference materials. The correctness of the method was checked in the entire measuring range.

#### 4.5.8. Extended Uncertainty

The expanded uncertainty U(x) was expressed as an exponential function depending on the silicon content. In this regard, components of uncertainty (coefficients of variation and relative errors) in the entire measuring range were collected. The uncertainty of a measurement procedure consists of the uncertainty of sampling and that of the analytical process. The expanded uncertainty is determined by multiplying the value of the composite standard uncertainty u(x) by the coverage factor k = 2 [35] according to the formula:(2)U(x)=k⋅u(x)

The complex uncertainty was determined in line with the law of propagation of standard uncertainties. In order to identify all components of uncertainty, an uncertainty budget was created. The uncertainty of the determination of silicon by the ICP-OES method is influenced by the initial sample preparation (drying, grinding, averaging), weighing the sample, sample mineralization, the transfer of the mineralized sample to a volumetric flask and the measurement of the silicon content.

## 5. Conclusions

A new, faster and safer methodological approach for silicon determination by ICP-OES technique preceded by microwave mineralization of samples has been developed and validated. The proposed method can be successfully used for the determination of silicon in routine analyses of zinc and lead-bearing materials and in waste. The determination of silicon in zinc and lead-bearing materials (i.e., zinc oxide and sulfide zinc and lead concentrates, zinc-lead ores and waste) using the developed method was implemented for routine analyses at the ZGH “Bolesław” laboratory.

## Figures and Tables

**Figure 1 molecules-27-03059-f001:**
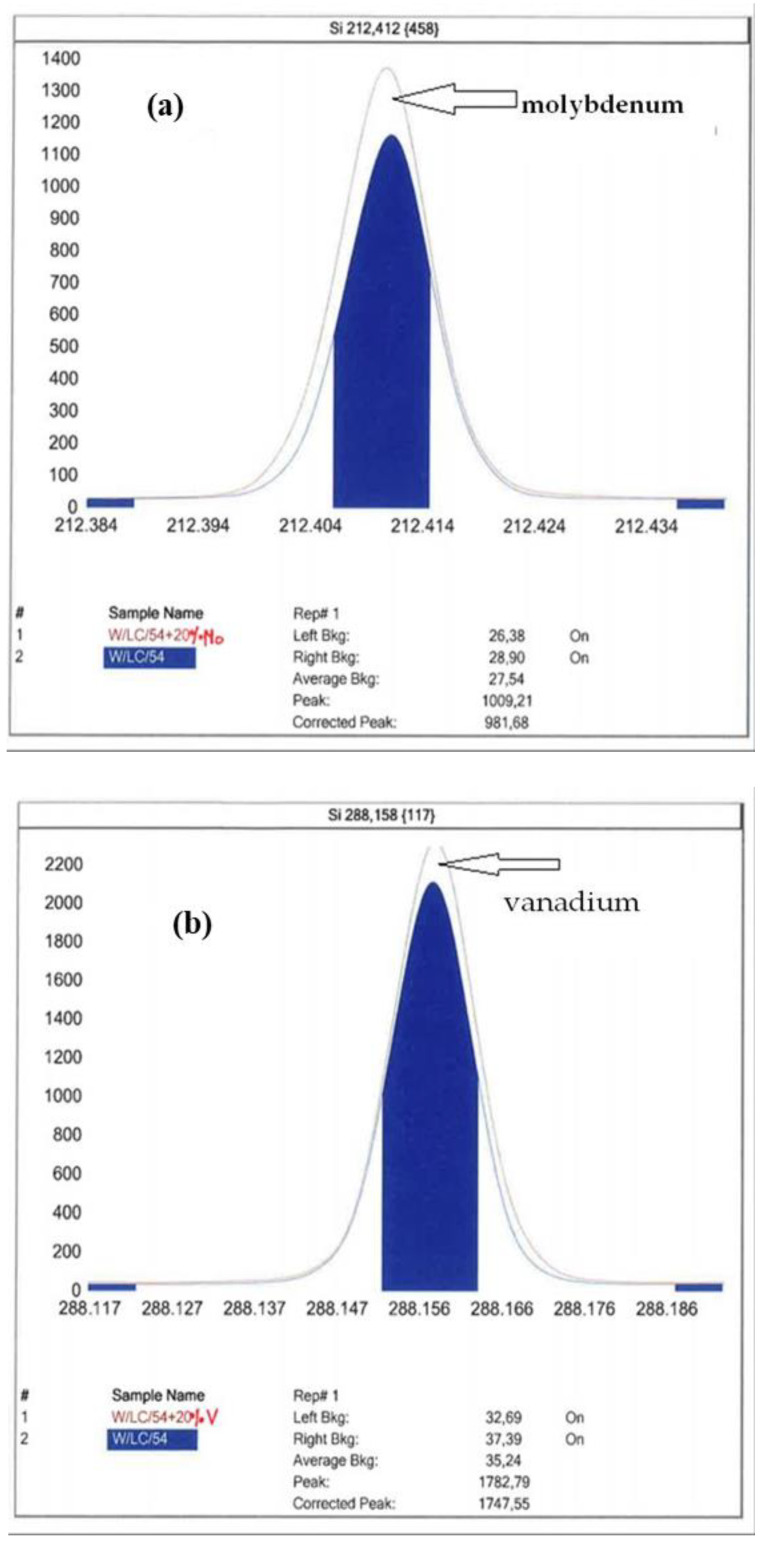
Silicon peak for CRM4 with the addition of 20% (**a**) molybdenum, (**b**) vanadium and (**c**) chromium.

**Figure 2 molecules-27-03059-f002:**
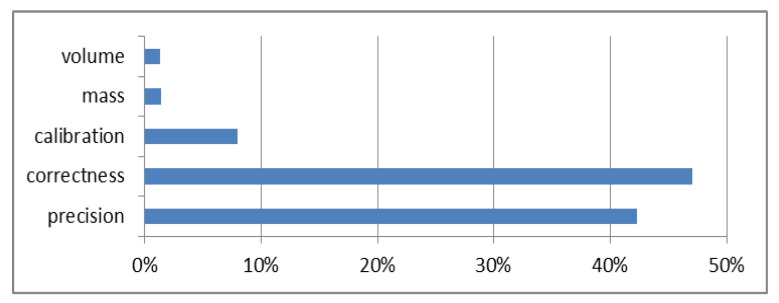
Percentage of components in the uncertainty budget for a silicon content of 0.1–10%.

**Figure 3 molecules-27-03059-f003:**
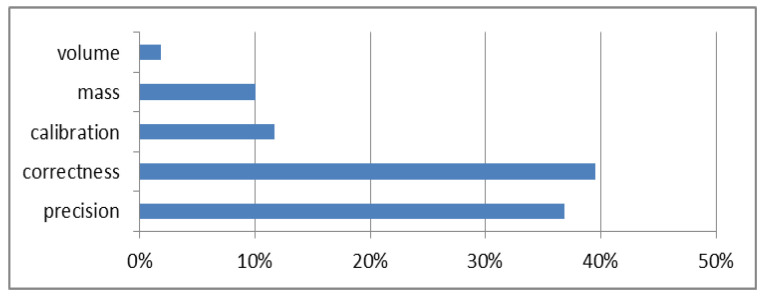
Percentage of components in the uncertainty budget for a silicon content of 10–50%.

**Figure 4 molecules-27-03059-f004:**
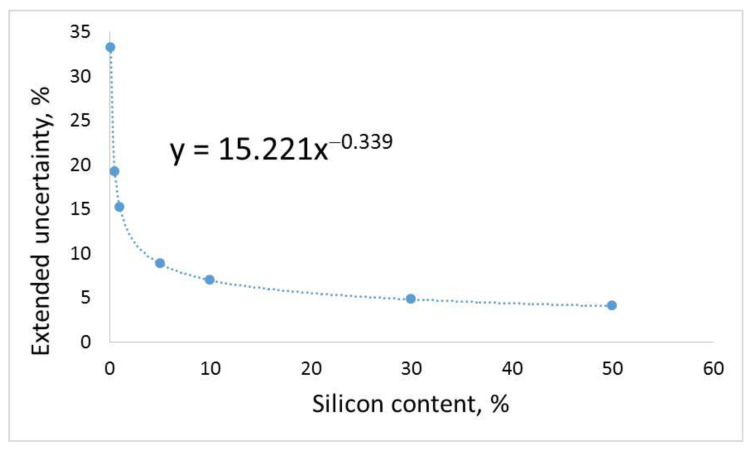
Expanded uncertainty as a function of silicon concentration.

**Table 1 molecules-27-03059-t001:** ICP-OES working parameters.

Parameter	Value
Radio frequency power (RF)	1150 W
Gas flow in the nebulizer	0.50 L/min
Auxiliary gas flow	0.5 L/min
Plasma gas flow	12 L/min
Pump speed	50 rpm
Purge flow	standard
Nebulizer pressure	210 a

**Table 2 molecules-27-03059-t002:** Comparison of the radial and axial ICP-OES systems.

Analytical Line, nm	RSD, %		LOQ, mg/L		Selectivity, 1/(mg/L) ^1^	
	Axial	Radial	Axial	Radial	Axial	Radial
212.412	0.12	0.13	0.012	0.013	1285	196
251.611	0.92	0.12	0.091	0.012	4915	583
288.158	0.29	0.16	0.029	0.016	3188	374

^1^ change in counts resulting in a 1 mg/L change in silicon concentration.

**Table 3 molecules-27-03059-t003:** Summary of validation parameters.

Parameter	Criteria	Results
Working range	0.10–50%	0.10–50%
Linearity	r ≥ 0.999	r = 1
LOD	LOD ≤ 0.05%	LOD = 0.050%
LOQ	LOQ ≤ 0.10%	LOQ = 0.10%
Selectivity	Interference may not significantly affect the test results	Molybdenum, chromium and vanadium interference does not statistically significantly affect the test results
Precision	1. RSD < 15% in the range of silicon content of 0.10–0.50 % 2. RSD < 10% in the content of silicon > 0.51%	1. Silicon content range of 0.10–0.50%: RSD_max_ = 10.2% 2. Silicon content > 0.51% RSD_max_ = 6.0%
Correctness	90% ≤ recovery ≤ 110%	90.0–101.1%
Extended uncertainty	1. U(x) < 50% in the silicon content range of 0.10–0.50% 2. U(x) < 30% in the silicon content > 0.51%	1. Silicon content range of 0.10–0.50%: U(x)max = 33.2% 2. Silicon content > 0.51%: U(x)max = 19.1%

**Table 4 molecules-27-03059-t004:** Silicon content with expanded uncertainty for certified reference materials.

Certified Reference Material	Silicon Content Determined ± U(x), %	Silicon Content Given by the Manufacturer ± U(x), %
CRM_1	0.110 ± 0.035	0.122 ± 0.028
CRM_2	0.265 ± 0.063	0.295 ± 0.019
CRM_3	0.303 ± 0.069	0.305 ± 0.029
CRM_4	2.59 ± 0.29	2.56 ± 0.11
CRM_5	9.05 ± 0.65	9.30 ± 0.06
CRM_6	38.02 ± 1.69	38.77 ± 0.10

**Table 5 molecules-27-03059-t005:** Silicon content in test samples.

Test Samples	Silicon Content ± U(x), %
1	0.190 ± 0.051
2	0.310 ± 0.070
3	0.550 ± 0.103
4	1.65 ± 0.21
5	2.98 ± 0.31
6	5.35 ± 0.46
7	7.56 ± 0.58
8	12.11 ± 0.79
9	25.34 ± 1.29

**Table 6 molecules-27-03059-t006:** List of tested samples.

Sample No.	Compound	Origin	Average Content of Zn and Pb, %	Appearance
1	ZnO	obtained from zinc bearing waste (steel dust)	Zn ≈ 62; Pb ≈ 3	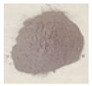
2	PbS	galena	Zn ≈ 3; Pb ≈ 63	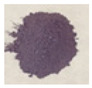
3	ZnO	obtained from zinc bearing waste (sludge)	Zn ≈ 57; Pb ≈ 2	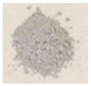
4	ZnS (blende)	imported from the Grot mine in Serbia	Zn ≈ 49; Pb ≈ 2.5	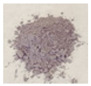
5	ZnS (blende)	imported from the Lece mine in Serbia	Zn ≈ 51; Pb ≈ 0.6	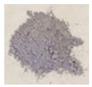
6	Zinc-bearing waste	sludge from flotation process (code 190205)	Zn ≈ 13; Pb ≈ 6	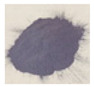
7	Zinc-bearing waste	Singen steel dust (code 100207)	Zn ≈ 35; Pb ≈ 0.1	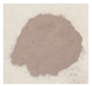
8	Zinc-bearing waste	Tiroler Rohre sludge (code 100213)	Zn ≈ 31; Pb ≈ 0.6	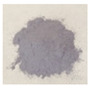
9	Zinc-lead ore	imported from Swedish mines	Zn ≈ 8; Pb ≈ 4	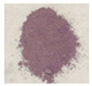

## Data Availability

Data is contained within the article.

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
