# Peer review of "A New Approach to the Determination of Silicon in Zinc, Lead-Bearing Materials and in Waste Using the ICP-OES Method"

_molecules, 2022, doi:10.3390/molecules27103059_

Round 1

Reviewer 1 Report

This manuscript reports the determination of silicon in zinc-lead materials and waste using an ICP-OES method.

The scientific background presented is solid and the authors clearly have experience developing analytical methods and writing analytical reports. The manuscript is clearly written and easy to follow.

My major concern is in the scientific originality of this work. To me what is presented is more a method development and validation based in a well-known analytical technique that has been previously used in many different samples as reported in the introduction. Also the importance of quantifying silicon in the samples measured (zinc-lead materials) is not clear in the manuscript. The feeling that I get from the lecture is that this is a niche interesting to a  small number of analytical laboratories. To be accepted as a scientific paper these two issues need to be clarified. 

The safety precautions during the mineralization (section 2.1 and 4.4) need to be discussed in more detail. The authors state that perchloric acid is eliminated from this methodology but they still use hydrofluoric acid which is one of the most dangerous chemical without any mention to safety precautions.

Another aspect that need to be discussed is the possible interference of glass parts from the ICP-OES instrument to the silicon measurement, especially in the low concentrations. Silicon is an element that ICP developers do not like due to this reason. In some instruments with lower detection limits, namely some ICP-MS, there are some special configurations to measure silicon where all glass parts are replaced by plastic parts in the instrument. This aspect needs to be discussed as well.

At this point the scientific novelty and interests of this works needs to be demonstrated and both the safety of the mineralization procedure and the possible interference from glass parts on the instrument needs to be adequately discussed so in my opinion this manuscript need a major revision. 

Author Response

Thank you very much for your review of the manuscript. We took into account your valuable comments and introduced additional descriptions to the text of the manuscript. The text with the marked changes is included in the attachment. We also thank you for paying attention to the aspect of the possibility of silicon interference in glass parts. In the instrument used, the atomiser is made of plastic, and at these silicon concentration levels, the remaining glass elements do not have a significant influence on the measurement results. Of course, we agree that if we want to carry out a trace level determination of silicon, we would have to use a more sensitive and suitable method, such as ICP-MS, which you have suggested. In addition, we have added information on the estimated production volume of these materials in Europe and the problems caused by excess silicon in these products. We hope that the introduced amendments and additions are in line with your suggestions. Kind regards, Authors.

Reviewer 2 Report

The study proposes presenting a new approach for determining silicon content in different samples through the ICP-OES technique. Despite being an interesting topic, the work has several flaws that, in my opinion, make it unpublishable in its current form. Some of them that concern me are described below:

a) Justification: the rationale for conducting the work, that is, the importance of the scientific problem is not well placed. In the Introduction, the authors briefly suggest that reducing spectral interferences are the main reason for the study, but it must be better developed. I would suggest including a basic scheme about the ICP-OES method, bearing in mind that non-specialists could also consult the article. Then, further information about current issues related to the technique, especially interferences, should be described and referenced in detail. In the next section (methodology), the differences between the standard and the new method should be highlighted. As far as I understood, such differences were limited to changing the dissolution medium (replacing sulfuric and perchloric acids with other solvents) and the analytical lines.

b) Advantages of the method: the advantage(s) of the proposed method over the common one(s) should be more well discussed. In the Conclusions, it is stated that "a new, faster, more effective and safer method...validated". Such statements should be grounded on comparative analysis to be included in the Discussion section.

c) Other issues:

  • The overall presentation is not good. In the submitted manuscript, the Results section was placed before the Methodology one. Some figures, like Figure 01, need better resolution for a good visualization.
  • An explanation must be provided about how the ICP-OES parameters defined in Table 02 were selected (manufacturer's manual?).
  • In section 4.4 it is suggested that while the mineralization and cooling time were kept constant, the initial heating profile of the sample was varied over the tests. In this case, these conditions of variation (heating rate, time step, etc) could be included.
  • The description of extended uncertainty in section 4.5.8 is confusing. I suggest rewriting this section to explain better how the parameter can be defined and calculated.

Author Response

Thank you very much for your review of the manuscript. We took into account your valuable comments. Additional descriptions and explanations are provided in the text (general description of the ICP-OES measurement principle in the Introduction section, description of the basic silicon determination method in the Discussion section, selection of ICP-OES parameters specified in Table 1). The text with the marked changes is included in the attachment. We also thank you for paying attention to the illegibility of the drawings. We have made some corrections. In addition, we have added information on the estimated production volume of these materials in Europe and the problems caused by excess silicon in these products. As for the general presentation of the manuscript, we wrote it based on the template provided by Molecules, in which after the introduction there is a section “Results” and “Discussion”, and then a description of the methodology. We hope that the introduced amendments and additions are in line with your suggestions. Kind regards, Authors.

Round 2

Reviewer 1 Report

It is acceptable.